# Potential of Bacterial Inoculants to Mitigate Soil Compaction Effects on *Gossypium hirsutum* Growth

**DOI:** 10.3390/plants14121844

**Published:** 2025-06-16

**Authors:** Fausto Henrique Viera Araújo, Crislaine Alves da Conceição, Adriene Caldeira Batista, Gabriel Faria Parreiras de Andrade, Caique Menezes de Abreu, Paulo Henrique Grazziotti, Ricardo Siqueira da Silva

**Affiliations:** Department of Agronomy, Federal University of the Jequitinhonha and Mucuri Valleys, Diamantina 39100-000, MG, Brazil; crislaine.alves@ufvjm.edu.br (C.A.d.C.); adriene.caldeira@ufvjm.edu.br (A.C.B.); gabriel.parreiras@ufvjm.edu.br (G.F.P.d.A.); menezes.abreu@ufvjm.edu.br (C.M.d.A.); paulo.grazziotti@ufvjm.edu.br (P.H.G.); ricardo.siqueira@ufvjm.edu.br (R.S.d.S.)

**Keywords:** biofertilizer, bioinputs, morphological parameters, nutrient availability, inoculants

## Abstract

Aims: Soil compaction is one of the main challenges in agriculture, negatively affecting cotton growth (*Gossypium hirsutum* L.), nutrition, and productivity. This study evaluated the efficacy of plant growth-promoting bacteria (PGPB), *Exiguobacterium sibiricum,* and *Pantoea vagans* in mitigating the effects of different soil compaction levels (65%, 75%, 85%, and 95%) on cotton performance. Methods: Parameters such as plant height, stem diameter, number of leaves, shoot dry matter (SDM), and nutrient content in leaves, stems, and roots were assessed. The methodology included variance analysis and mean clustering to identify significant differences among treatments using R software. Results: The results indicated that PGPB inoculation improved plant growth and nutrition even under high compaction levels. Cotton height increased by up to 45% in compacted soils (95%), while stem diameter and SDM also showed significant gains. Foliar nutrient levels of N (37.2 g kg^−1^), Ca, and Mg remained within the adequate range for cotton cultivation, reflecting the efficiency of PGPB in enhancing nutrient absorption. Under severe compaction, Ca accumulation dropped to 18.2 g kg^−1^, highlighting the physical constraints imposed on the roots; however, the bacterial action mitigated this impact. Additionally, bacterial strains increased the availability of N and P in the soil due to their ability to fix nitrogen, solubilize phosphates, and produce exopolysaccharides that improve soil structure. Conclusions: In conclusion, inoculation with *Exiguobacterium sibiricum* and *Pantoea vagans* is an effective strategy to mitigate the impacts of soil compaction on cotton. These bacteria promote plant growth and nutrition and enhance the soil’s physical and biological properties.

## 1. Introduction

The cotton plant (*Gossypium hirsutum* L.) is an agricultural crop cultivated in approximately 150 countries with an estimated planted area of 35 million hectares per year [1]. Cotton fiber is one of the primary raw materials driving the global economy, and it is widely used in clothing and household furnishings. Additionally, rich in oil and nutrients, cotton seeds have significant applications in the bioenergy and animal nutrition sectors [2,3].

However, the intensive and continuous cultivation of cotton, particularly in countries such as India, China, the United States, Pakistan, and Brazil, which together account for 70% of global production [4], has contributed to the physical degradation of soil. This degradation occurs primarily due to the heavy traffic of agricultural machinery in cultivated areas [5,6].

The alteration of soil physical attributes due to compaction increases particle density and reduces total pore volume [5,7]. This environmental issue compromises nutrient availability, water movement, and soil aeration, hindering the growth of cotton plants [2,8] and causing productivity losses of up to 27% [9].

Production reductions result from direct and indirect damage, which is primarily due to the inefficient use of water in subsurface layers caused by physical barriers to root development. In this context, it is essential to understand the factors limiting the early growth of cotton plants and to explore new biotechnologies that can mitigate the adverse effects of soil compaction.

The inoculation of plant growth-promoting bacteria (PGPB) has emerged as a promising technique to enhance plant performance under adverse cultivation conditions. These bacteria can be isolated from diverse habitats, function either as endophytes or free-living organisms [10] when in synergistic association with plants, can protect against edaphoclimatic variations [11].

Bacterial species from the genera *Bacillus*, *Pantoea vagans*, and *Exiguobacterium sibiricum* [12,13,14] interact with plants either generally or, in some cases, specifically. The beneficial effects of these bacteria are primarily attributed to the synthesis of phytohormones and enzymes [10], promoting crucial physiological responses such as the induction of resistance to abiotic and biotic stressors, stimulation of root growth, siderophore production, solubilization of inorganic nutrients, and synthesis of enzymes including nitrate reductase, nitrogenase, catalase, and urease [15].

These characteristics highlight the potential of PGPB in selecting specific strains for cotton (*Gossypium hirsutum*), stimulating root growth, and optimizing the physiological activity of plants, even in environments considered unsuitable. Thus, this study not only aims to evaluate the early development and macronutrient levels in *G. hirsutum* under soil compaction but also to analyze how *P. vagans* and *E. sibiricum* can promote growth and improve nutrient absorption efficiency under adverse cultivation conditions.

This study evaluated the early growth and macronutrient levels in *G. hirsutum* cultivated under soil compaction and inoculated with *P. vagans* and *E. sibiricum*.

## 2. Material and Methods

### 2.1. Experiment Planning and Installation

The experiment was conducted from December 2022 to February 2023 in a greenhouse oriented in a north–south direction, located at the Department of Agronomy of the Federal University of the Jequitinhonha and Mucuri Valleys (UFVJM), in Diamantina, State of Minas Gerais, Brazil (18°14′58″ S, 43°36′01″ W, at an altitude of 1113 m).

The experimental design was carried out in a randomized block design with four repetitions of a 2 × 4 factorial scheme. The first factor consisted of the joint inoculation (bacterial mix) of the endophytic bacteria *Exiguobacterium sibiricum* strain 19RP3L2-7 (GenBank code: OQ983562) and *Pantoea vagans* strain 7URP1-6 (GenBank code: OQ983564) compared to the non-inoculated condition. The second factor involved the following soil compaction levels: 65%, 75%, 85%, and 95%.

The experimental unit consisted of a cotton plant (*Gossypium hirsutum* L.) of the BRS500 B2RF cultivar, grown in a polyvinyl chloride (PVC) column with a height of 40 cm, divided into four overlapping rings of 10 cm in height and 10 cm in diameter. The three lower rings were compacted according to the soil compaction levels, while the top ring was not compacted, simulating conventional soil preparation. The fertilization followed the nutritional recommendations for cotton [16].

### 2.2. Substrate Used and Normal Proctor Test

The soil used for plant cultivation was a Ferralsol [17], classified as Red Latosol [18], sieved through a 4 mm mesh. The chemical and physical composition of the soil presented the following characteristics: pH (H_2_O) = 5.93; P (remaining phosphorus) = 3.10 mg dm^−3^; K⁺ (potassium) = 25.05 mg dm^−3^; Ca^2^⁺ (calcium) = 2.54 cmolc dm^−3^; Mg^2^⁺ (magnesium)= 0.77 cmolc dm^−3^; Al^3^⁺ (aluminum)= 0.09 cmolc dm^−3^; H + Al (potential acidity) = 3.41 cmolc dm^−3^; SB (sum of bases) = 3.37 cmolc dm^−3^; t (effective cation exchange capacity) = 3.46 cmolc dm^−3^; T (cation exchange capacity at pH 7.0) = 6.78 cmolc dm^−3^; V (base saturation) = 50%; m (aluminum saturation) = 3%; OM (organic matter) = 1.94 dag kg^−1^; sand = 42 dag kg^−1^; silt = 14 dag kg^−1^; clay = 44 dag kg^−1^. The analyses of P and K were performed using the Mehlich-1 extractor, of Ca^2^⁺, Mg^2^⁺, and Al^3^⁺ using 1 mol L^−1^ KCl extractor, of H + Al using 0.5 mol L^−1^ calcium acetate extractor, and organic matter (OM) was determined by the carbon oxidation method with potassium dichromate in an acidic medium.

The regular Proctor test determined the maximum soil density and optimal compaction moisture [19,20,21,22]. In this test, a soil sample was compacted within a cylinder of approximately 1000 cm^3^ volume in three successive layers with 25 blows applied using a 2.5 kg rammer from a height of 30 cm. The test was repeated for different moisture levels until six samples of the same volume were obtained, ranging from moist to extremely moist soil. Each sample was weighed, and a portion was retained to determine the moisture content. The natural density of the collected soil was determined using the volumetric ring method [23].

The results were plotted in a graph (gravimetric moisture versus soil density) and fitted by a second-degree polynomial (Ds = aU^2^ + bU + c), obtaining the compaction curve, with the optimal humidity (Uₒₜ) of 19% and the maximum soil density (Dₛₘₐₓ) of 1.54 g cm^−3^, which was calculated using the following expressions: Uₒₜ = −b/2a; Dₛₘₐₓ = −(b^2^ − 4ac)/4ª, where a, b, and c are the parameters of the equation.

To achieve the compaction levels of 65%, 75%, 85%, and 95%, the ratio between the natural soil density and the maximum density was calculated as well as the soil mass (kg) using the formula (CG = (Ds/Dsm) × 100), where CG = Compaction Level, Ds = Natural Soil Density, and Dsm = Maximum Soil Compaction Density [24]. Knowing the desired compaction level (65%, 75%, 85%, or 95%) and the volume of the experimental container (PVC cylindrical rings in cm^3^), it is possible to calculate the soil mass (kg) needed to achieve the desired compaction level.

### 2.3. Obtaining Microbial Accessions and Bacterial Production and Inoculation

The bacteria used in this study were obtained from the Soil Microbiology Laboratory at Federal University of the Jequitinhonha and Mucuri Valleys (UFVJM). The cells of *E. sibiricum* strain 19RP3L2-7 were isolated from the roots of adult *E. camaldulensis* plants, showing the production of indole-3-acetic acid (IAA: 90 μg mL^−1^), catalase, nitrate reductase, and phosphatase. *P. vagans* strain 7URP1-6 was isolated from seedlings of *Eucalyptus grandis* × *Eucalyptus urophylla*, with IAA production (129 μg mL^−1^), catalase, nitrogenase, urease, nitrate reductase, and phosphatase production [25].

For the preparation of the inoculant, bacterial cells were grown in liquid Luria–Bertani medium [26] and incubated at 28 °C for 72 h under orbital shaking at 120 rpm (Shaker Incubator Cienlab, São Paulo, Brazil). Bacterial cells of each strain were incubated separately in vitro. After incubation, the cells were centrifuged at 4000 rpm for 10 min (centrifuge Kasvi K14-400) and resuspended in 0.85% saline solution [27]. The inoculant was prepared with the polymer carboxymethylcellulose (60%) and starch (40%) (*w*/*v*) at a concentration of 0.8 g L^−1^ in sterilized distilled water (patent: PI0506338-8). We determined the concentration of 10⁹ cells/mL of bacterial cells per milliliter for the inoculant calibration of each bacterial strain [28]. The bacterial inoculation was performed once at planting and applied directly to the cotton seeds using the liquid medium containing carboxymethylcellulose at a dose of 5 mL per seed.

### 2.4. Assessments of Vegetative Growth Promotion

At 70 days after plant emergence, the growth of *Gossypium hirsutum* was assessed by measuring plant height (cm), stem diameter (mm), the number of lateral branches, the number of leaves, and the dry matter of the shoot (SDM) and roots (g). Additionally, the number of colony-forming units (CFU g^−1^ of tissue) of the bacteria in the rhizosphere, roots, and leaves was evaluated. Subsequently, a nutritional analysis of macronutrients, nitrogen (N), phosphorus (P), potassium (K), calcium (Ca), and magnesium (Mg), was conducted for the leaf, stem, and root portions of the plant.

#### 2.4.1. Determination of the Nutritional Content of Plants

The leaf and root samples were dehydrated in a forced-air circulation oven (Solab SL100; Solab Scientific Laboratories Equipment, Piracicaba, Brazil) at 65 °C until the material reached a constant weight. The samples were then ground using a knife mill (Willey type, TE-650; Solab Scientific Laboratories Equipment, Piracicaba, Brazil), homogenized, and stored. A subsample was digested in nitric acid (HNO₃) using a closed-system microwave oven (CEM: Mars XPRESS; CEM Corporation, Matthews, North Carolina, USA) [29], and the product was diluted to a 1:1 concentration. Phosphorus (P) content was determined using the spectrophotometric method with molybdenum blue (brand: model) [30]. The K, Ca, and Mg were measured using atomic absorption spectrophotometry (Shimadzu: AA-7800; Shimadzu Corporation, Kyoto, Japan) [30,31]. Nitrogen content was determined using an elemental analyzer (TruSpec Micro CHN, LECO; LECO Corporation, St. Joseph, MI, USA).

#### 2.4.2. Microbial Density in Plant Tissues and Rhizosphere

Colony-forming units (CFU) were determined to estimate the concentration of bacterial cells per gram of plant tissue. One plant per treatment was randomly selected and then segmented into leaves, roots, and rhizosphere (substrate adhered to the roots). For root disinfestation, the substrate was manually removed, and the roots were placed in saline solution (9 mL of saline solution per 1 g of roots) and orbitally shaken for 30 min at 200 rpm to displace the bacteria from the rhizosphere [32]. The surface disinfection of each plant segment was conducted as described by [32].

To obtain the rhizosphere solution, 1 g of roots was placed in 9 mL saline solution and agitated at 200 rpm for 30 min (Shaker Incubator Cienlab, São Paulo, Brazil), following [27]. For the preparation of leaf and root tissue suspensions, the disinfected material was macerated at a 1:9 ratio (tissue: saline solution), which was followed by agitation at 200 rpm for 30 min using the same equipment [27]. The resulting suspensions were subjected to serial dilutions and standardized at concentrations of 10⁻⁵ for leaves, 10⁻⁶ for roots, and 10⁻⁹ for the rhizosphere.

For CFU/g^−1^ determination, 0.1 mL of the suspension was transferred to Petri dishes containing tryptone soy agar (TSA) medium, spread using the plating method, and incubated in a bacteriological chamber (Solab Sb224, São Paulo, Brazil) at 28 °C for 24 h. Three replicates per treatment were used, and colonies were counted within 200–300 colonies per plate [33].

### 2.5. Statistical Analysis

The data distribution was analyzed using the Durbin–Watson test (independence of errors), the Shapiro–Wilk test (normality), and Bartlett’s test (homogeneity of variances). After verifying these assumptions, analysis of variance (ANOVA) was conducted. When significant effects were detected, the means were grouped using the Scott–Knott test, employing the RStudio software, version R 4.3.3 [34].

## 3. Result

An increase in the density of colony-forming units (CFU/g) in the rhizosphere was observed with higher levels of soil compaction. In the inoculated experimental units, rhizospheric activity resulted in estimated microbial populations ranging from 2.0 × 10^12^ to 3.0 × 10^12^ CFU/g. The quantification of the microbial population associated with the roots revealed variations in the density of CFU/g, ranging from 1.0 × 10^8^ in non-inoculated plants to 3.0 × 10^10^ in inoculated ones, with the highest densities observed under 85% and 95% soil compaction.

The shoot dry matter (SDM), plant height (cm), N and K contents (g kg^−1^) in leaves, and N content (g kg^−1^) in stems were influenced by the interaction between inoculation of the bacterial mix *Pantoea vagans* strain 7URP1-6 and *Exiguobacterium sibiricum* strain 19RP3L2-7 and soil compaction levels (*p* < 0.05). In the nutritional analysis, a standalone effect of soil compaction was also observed on Ca and Mg contents (g kg^−1^) in leaves, P content (g kg^−1^) in stems, and N content in roots (*p* < 0.05).

A significant effect of bacterial inoculation was detected on N and Mg contents in roots (*p* < 0.05). For the variables number of leaves, lateral branches, and stem diameter (mm), only the simple effects of both treatments were identified (*p* < 0.05). Conversely, for root dry matter (g) and colony-forming units, the statistical assumptions were not met, or no significant influence of bacterial inoculation or soil compaction was observed (*p* < 0.05).

### 3.1. Initial Growth of Cotton Plant (Gossypium hirsutum L.) 

The growth of cotton plants inoculated with plant growth-promoting bacteria (PGPB) showed increases of 10%, 32%, and 25% in the stem diameter (mm), number of lateral branches, and number of leaves, respectively, compared to treatments without inoculation (Figure 1A and Appendix A). In the absence of inoculants and under high soil compaction (95%), the number of lateral branches decreased by 18% and the number of leaves by 30% compared to soil compaction levels of 65%, 75%, and 85% (Figure 1B).

Plant height (cm) averages were statistically similar in treatments with PGPB inoculation regardless of soil compaction levels. However, in treatments without inoculation, 95% compaction resulted in a 25% reduction in plant height (Figure 1C and Appendix A). Furthermore, there was an 8% and 45% increase in plant height with PGPB inoculation at soil compaction levels of 65% and 95%, respectively, compared to non-inoculated treatments.

Shoot dry matter (SDM, g) was significantly influenced by the interaction between PGPB inoculation and soil compaction levels. Inoculated plants subjected to 75%, 85%, and 95% compaction levels exhibited 47% lower SDM accumulation than at the optimal compaction of 65%. In non-inoculated treatments, 95% compaction led to a 33% reduction in SDM accumulation compared to other compaction levels (Figure 1D). Additionally, a 59% increase in SDM accumulation was observed in inoculated plants at 65% compaction compared to non-inoculated treatments.

### 3.2. Macronutrient Contents in Plant Tissues

#### 3.2.1. Leaf

The N content in the leaves was higher in the inoculated plants (37.2 g kg^−1^), showing a 27% increase in N accumulation compared to the non-inoculated plants at 65% soil compaction (Figure 2B). As soil compaction increased (85% and 95%), N accumulation decreased by approximately 25% compared to the 65% and 75% compaction levels. Regarding the leaves’ P content, no significant differences were observed between treatments, although P levels varied around 2 g kg^−1^.

The potassium content in the leaves was higher in the inoculated plants (36.6 g kg^−1^), with a 16% increase in K accumulation, especially in plants grown at 65% compaction (Figure 2C). As compaction increased (95%), K accumulation (30.2 g kg^−1^) decreased by approximately 17% compared to the 65% and 75% compaction levels (Figure 2C). No significant differences were observed in non-inoculated treatments.

The magnesium content in the leaves (4 g kg^−1^) in plants grown in soil with 85% and 95% compaction decreased by 32% compared to the 65% compaction level (5.9 g kg^−1^) (Figure 2A). Similarly, the Ca content in plants with 85% and 95% compaction decreased by 20% and 48%, respectively, compared to the 65% compaction level (34.3 g kg^−1^) (Figure 2A).

#### 3.2.2. Stem

For the N content in the stem (g kg^−1^), the plants inoculated with BPC and subjected to soil compaction stress (75%, 85%, and 95%) showed lower N accumulation with a decrease of approximately 18% compared to the optimal compaction condition of 65% (21 g kg^−1^). In the non-inoculated treatment, when plants were subjected to 95% compaction stress (14 g kg^−1^), a 22% reduction in N accumulation was observed compared to the 75% compaction level (Figure 2E). A higher accumulation of N in the stems was observed in the inoculated plants with an increase of 36% and 25% when exposed to 65% and 95% compaction levels, respectively, compared to the non-inoculated plants (Figure 2E).

Regarding P content in the stem, a statistical difference was observed based on the soil compaction treatments. When exposed to 85% compaction stress (1.89 g kg^−1^) and 95% compaction (2 g kg^−1^), the plants exhibited approximately 17% lower P accumulation compared to the 65% compaction treatment (2.2 g kg^−1^) (Figure 2D).

For K content in the stem, the inoculated plants accumulated higher levels of K (43.3 g kg^−1^) compared to the non-inoculated treatment (35.3 g kg^−1^) at 65% compaction (Figure 2F). For Ca and Mg in the stem, the treatments either did not meet the statistical assumptions or no significant influence of bacterial inoculation or soil compaction was observed. However, the average concentrations of these elements varied around 35 g kg^−1^ for Ca and 5 g kg^−1^ for Mg.

#### 3.2.3. Roots

For the N content in the root, a statistical difference was observed based on the soil compaction treatments. When subjected to compaction stress at 75%, 85%, and 95%, the plants showed lower N accumulation with a decrease of approximately 15% (15.2 g kg^−1^) compared to the 65% compaction level (17.9 g kg^−1^) (Figure 2G). Additionally, a higher accumulation of N and Mg was observed with increases of 21% and 10%, respectively, in the plants inoculated with BPC compared to the non-inoculated plants, reaching 17.2 g kg^−1^ for N and 2.6 g kg^−1^ for Mg (Figure 2H). The treatments did not meet the root’s statistical assumptions for P, K, and Ca. However, the average concentrations of these elements ranged around 2.4 g kg^−1^ for P, 35.4 g kg^−1^ for K, and 3.7 g kg^−1^ for Ca.

## 4. Discussion

Soil compaction (85% and 95%) promoted an increase in the density of colony-forming units (CFU/g) in the rhizosphere, specifically in the inoculated experimental units. The bacteria *P. vagans* and *E. sibiricum* are producers of extracellular polysaccharides (EPS), which can aid in the aggregation of soil particles, restructuring, and the reduction in compaction in the rhizospheric region of plants. This enables an increase in enzymatic activity for nutrient solubilization and stimulates the production of plant hormones and organic acids [35]. The anions from the exuded organic acids can selectively favor microorganisms associated with macronutrient acquisition during the early stages of plant growth in compacted soils [36].

The *E. sibiricum* and *P. vagans* have been identified as generalist microorganisms capable of promoting the growth of plants such as *Eucalyptus* sp., *Corymbia* sp., and *Fragaria* × *ananassa* [32,37] with emphasis on the development of the root system. This effect was observed in plants subjected to micropropagation via cuttings and seeds and was mediated by the synthesis of phytohormones, nitrogen fixation, siderophore production, solubilization of inorganic nutrient sources, and antagonism to phytopathogens [18].

The *E. sibiricum* is capable of synthesizing the enzyme nitrate reductase, and it is responsible for nitrogen assimilation in the form of NH₄⁺. The *P. vagans*, in turn, exhibits the ability for biological nitrogen fixation and the synthesis of enzymes such as urease and nitrate reductase, as well as the solubilization of inorganic and organic phosphorus sources such as calcium phosphate and sodium phytate [25,32]. The strains used have characteristics that favor plant growth, such as the production of indole-3-acetic acid (IAA), siderophores, phosphate solubilization, ammonia release, hydrogen cyanide production, and exopolysaccharide synthesis [14,38]. Although the selection of bacteria for inoculant formulation is based on their production of secondary metabolites, their actual function is determined by the microorganism–plant–environment interaction.

The physical conditions of the soil required for optimal cotton (*Gossypium hirsutum* L.) performance involve a density of approximately 1.2 to 1.5 g cm^3^ and porosity between 47.2% and 54.5%. Values above these may hinder plant growth and development [39]. Soil compaction induces plants to increase the production and diversity of root exudates, facilitating root penetration, soil restructuring, and interaction with beneficial microorganisms. These adaptations are essential for the survival and productivity of *G. hirsutum* plants in response to dense structure and limited aeration.

Inoculation with PGPB stimulates enzymatic activity in the rhizosphere, enhancing nutrient solubilization, the production of plant hormones and organic acids, all of which are frequently associated with root exudates. Furthermore, it is noteworthy that modifications in the rhizospheric microbiota may directly influence the composition of exudates, although this specific mechanism was not detailed in this study.

Cotton growth, when inoculated with PGPB, is favored even in soils with high levels of compaction (65%, 75%, 85%, and 95%) (Figure 1A and Appendix A). Without inoculation and under 95% compaction, there was an 18% reduction in the number of lateral branches and a 30% decrease in the number of leaves compared to lower levels of compaction (Figure 1B). This indicates that the presence of microorganisms contributes to plant development under adverse soil conditions. The interaction between soil texture and nutrient levels creates a dynamic environment that shapes the microbial community structure, in which nutrient availability plays a fundamental role in the formation of exudate profiles, such as organic acids, which recruit beneficial microbes and assist in nutrient acquisition and disease suppression [40].

Inoculation with *P. vagans* may have promoted the production of biosurfactants, altering water surface tension and facilitating nutrient and water movement in the soil [41], which contributes to increased resilience to water stress. An increase of up to 45% was observed in plant height, as well as in stem diameter and shoot dry matter (SDM), suggesting that the inoculant is effective even in compacted soils (Figure 1A,C and Appendix A). Furthermore, non-inoculated plants grown in highly compacted soils (75%, 85%, 95%) presented 33% lower SDM (Figure 1D). This highlights the impact of soil compaction in agricultural areas and the difficulty faced by the native microbiota in promoting soil restructuring, resulting in reduced stability for the growth, development, and production of cotton (*Gossypium hirsutum*) [42]. Similar results were observed by other researchers, such as [43], who reported a 38% reduction in SDM in areas with compaction above 55%.

Foliar levels of nutrients such as Ca, Mg, and N were higher in inoculated plants, reflecting improved absorption and nutrition even under severe compaction conditions. Nitrogen is essential for processes such as energy transfer (ATP), plant growth, and leaf area development, all of which are crucial for stress resistance [44,45]. The Ca and Mg levels were within the range considered adequate for cotton cultivation, varying from 20 to 35 g kg^−1^ for Ca and 3 to 8 g kg^−1^ for Mg, reflecting improvements in soil properties and nutrient uptake efficiency promoted by the bacteria [30,46]. The low Ca accumulation (18.2 g kg^−1^) in soils with 95% compaction is closely related to the physical and biological limitations of root absorption, affecting the transport and retention of the element in leaves.

Higher nitrogen contents (g kg^−1^) were observed in plants inoculated with bacterial strains compared to non-inoculated treatments (Figure 2B). The foliar N content (37.2 g kg^−1^) in the PGPB treatment under 65% soil compaction falls within the sufficiency range for cotton, which is 35 to 43 g kg^−1^ [30,46]. Nitrogen is an essential element for the formation of enzymes, alkaloids, nucleic acids, and proteins, constituting approximately 16% of plant protein biomass [45]. Additionally, it plays a fundamental role in photosynthetic processes, participating in energy transfer (ADP and ATP), promoting plant growth, leaf area expansion, and consequently, increased productivity [44]. The innovative aspect of this study lies in the use of microorganisms as a biotechnological tool to address the agronomic challenge of soil compaction, which is a critical abiotic stress that reduces global agricultural productivity. PGPB contributed not only to plant nutrition but also to improvements in soil structure and microbiota, enabling plant growth under adverse conditions.

Regarding average leaf potassium content (Figure 2C), both inoculated (33.6 g kg^−1^) and non-inoculated treatments (31.7 g kg^−1^) exceeded the reference range for cotton, which is 15 to 25 g kg^−1^ [30,46]. This trend was also observed in potassium accumulation in the stem.

For nitrogen and phosphorus contents in the stem (g kg^−1^), higher accumulation values were recorded under PGPB inoculation at lower soil compaction levels (Figure 2D,E). However, the phosphorus content (2.4 g kg^−1^) was below the sufficiency range for cotton, which is 2.5 to 4 g kg^−1^ [30,46]. Microbial activity in the soil promotes the increased fertility and accessibility of nutrients to plants [47].

Strains of the Exiguobacterium genus exhibit characteristics that favor plant growth, including the production of indole-3-acetic acid (IAA), siderophores, phosphate solubilization, ammonia release, hydrogen cyanide production, and exopolysaccharide synthesis [14,38].

These data indicate that soil compaction influences the microbial community in the rhizosphere, increasing the concentration of microorganisms in this region. Furthermore, bacterial inoculation significantly increases root surface colonization, which is evidenced by an increase of up to two orders of magnitude in CFU/g density compared to non-inoculated plants. The observation of higher microbial densities in the roots of inoculated plants under high compaction suggests a synergistic effect between inoculation and the stress conditions imposed by compaction, enhancing microbial association with the root system.

Despite the promising results of inoculation with *Exiguobacterium sibiricum* and *Pantoea vagans* in mitigating the effects of soil compaction on cotton—promoting growth, nutrition, and improvements in soil properties, the study has some limitations. One of these is the lack of detailed information on the activity of root exudates and their interaction with other microorganisms, suggesting the need for broader analyses such as metagenomic sequencing. Additionally, the efficacy of the strains needs to be evaluated in different soil types and climatic conditions as well as the long-term impact of inoculations.

The findings of this study offer valuable insights into more sustainable agricultural practices, demonstrating the potential of inoculation as a strategy to address soil compaction. However, further field validations are essential to confirm the large-scale applicability of these strategies in real-world agricultural scenarios.

## 5. Conclusions

Soil compaction increases microbial density in the rhizosphere in *Gossypium hirsutum*.

Soil compaction reduced the initial growth and levels of Ca, Mg (leaf), P (stem), and N (root) in *Gossypium hirsutum*.

Inoculation with *Pantoea vagans* strain 7URP1-6 and *Exiguobacterium sibiricum* strain 19RP3L2-7 resulted in higher dry matter accumulation and increased foliar levels of N and K.

Using these bacterial inoculants appears promising for optimizing the growth of *Gossypium hirsutum* both under optimal conditions and in compacted soils.

## Figures and Tables

**Figure 1 plants-14-01844-f001:**
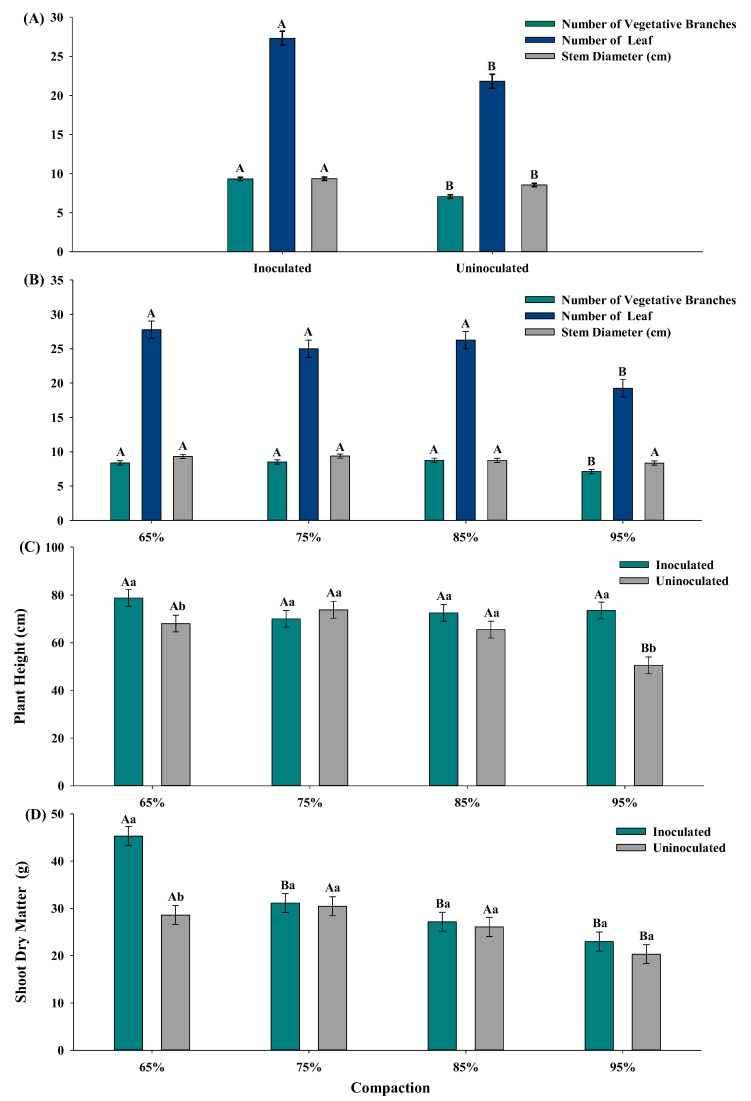
(**A**,**B**) Stem diameter (mm), number of lateral branches, and leaves. (**C**) Plant height (cm). (**D**) Shoot dry matter (g) under different soil compaction levels and inoculated with *Pantoea vagans* strain 7URP1-6 and *Exiguobacterium sibiricum* strain 19RP3L2-7. Means followed by the same uppercase letter in the column do not differ according to the Skott–Knott test (*p* < 0.05). Means followed by the same lowercase letter in the column between inoculated and non-inoculated treatments do not differ according to the Skott–Knott test (*p* < 0.05).

**Figure 2 plants-14-01844-f002:**
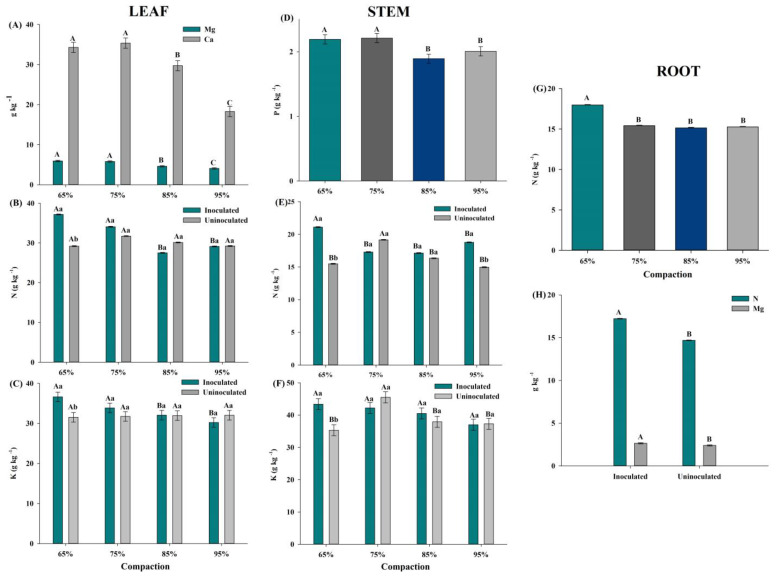
(**A**–**C**) Mg, Ca, N, and K (g kg^−1^) in the leaf. (**D**–**F**) P, N, and K (g kg^−1^) in the stem. (**G** and **H**) N and Mg in the root (g kg^−1^) of cotton plants under different soil compaction levels and inoculated with *Pantoea vagans* strain 7URP1-6 and *Exiguobacterium sibiricum* strain 19RP3L2-7. Means followed by the same uppercase letter in the column do not differ according to the Skott–Knott test (*p* < 0.05). Means followed by the same lowercase letter in the column between inoculated and non-inoculated treatments do not differ according to the Skott–Knott test (*p* < 0.05).

## Data Availability

The original contributions are included in the article, and Appendix A. Further queries can be directed at the corresponding author.

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
