# Peer review of "Potential of Bacterial Inoculants to Mitigate Soil Compaction Effects on Gossypium hirsutum Growth"

_plants, 2025, doi:10.3390/plants14121844_

Round 1
Reviewer 1 Report
Comments and Suggestions for Authors
This is an interesting and useful article showing capacity of some bacterial inoculants to mitigate effects of soil compaction on the growth and accumulation of mineral elements in cotton. Still I have some remarks. They should be addressed before I can recommend publishing this article.
1. Two bacterial strains (Exiguobacterium sibiricum strain 19RP3L2-7 and Pantoea vagans strain 7URP1-6) were used for inoculation in the present work. Since it is not specified in the result section, which strains were used for inoculation, it is can be assumed that their mixture was used for plants treatment. I strongly recommend to make this clearer and to explain, if bacterial strains were incubated separately in vitro and in which proportion the cells of these strains were combined for the seed treatment.
2. “(CG = (Ds / Dsm) × 100), where CN = Compaction Level, Ds = Natural 122 Soil Density, and Dsm = Maximum Soil Compaction Density” - Should it be CG or CN? CG is not deciphered, while CN is absent from the formula.
3. Citation in the text should be arranged according to the guidelines of “Plants”. I recommend to download any recent article and to check this. It is said in the guidelines that “In the text, reference numbers should be placed in square brackets.” So in the text there should be numbers and not Family et al. in the brackets.
4. One more remark concerning arrangement of the literature list. It should be checked and names of the authors presented in the same way. In the present state it is Oliveira in one case and de Oliveira – in the other.
- “One plant per treatment was randomly selected and then segmented into leaves, roots, and rhizosphere (substrate adhered to the roots). Surface disinfestation of each plant segment was conducted as described by Oliveira et al. (2024)” – when I read this sentence, I failed to understand why plants were sterilized before assay of CFU in rhizosphere. But then I read the article of Oliveira et al., (2024) and found that “Before root disinfestation, the substrate was removed manually and the roots were placed in saline solution (9 mL of saline solution per 1 g of roots) and agitated orbitally for 30 min at 200 rpm to displace the bacteria from the rhizosphere”. So disinfection was performed after bacteria from root rhizopshere were recovered. I believe the authors should point this out rather than relying solely on the reference to Oliveira et al, (2024).
- Furthermore, I recommend to modify the title of section 2.4.2. by adding rhizosphere: Microbial Density in Plant Tissues and Rhizosphere.
- I failed to find exact information about microbial density either in plant tissues or rhizospeher. It is only mentioned that “for colony-forming units, the statistical assumptions were not met”. Why were methods of their determination described in such details in M & M section?
- Root characteristics under different level of soil compaction and inoculation are also not shown. Deficiency of these data is possibly due to the observation that “for root dry matter … the statistical assumptions were not met, or no significant influence of bacterial inoculation or soil compaction was observed.” Still I recommend providing this data as supplementary material
- “Shoot dry matter (SDM, g) was significantly influenced by the interaction between PGPB inoculation and soil compaction levels. Inoculated plants subjected to 75%, 85%, 216 and 95% compaction levels exhibited 47% lower SDM accumulation than at the optimal 217 compaction of 65%.” – the statement about interaction of the factors should be supported by two-way ANOVA. Its result should be presented as a table showing F, p and Fcrit values.
- I recommend to start Discussion from results obtained in the present work and then to relate them to the literature data. Otherwise too many literature data are discussed, which are not related to what was measured in the present work. For example authors mention that “This effect may be related to reduced exploration of the cotton plant's root system, resulting in decreased water and nutrient absorption.” But no data on root dimensions are shown in the present work.
- Throughout the text authors mention “treatments inoculated with bacteria”. I think it should not be forgotten that not treatments, but plants are inoculated. I recommend modifying such phrases throughout the text.
- “The strain Exiguobacterium sibiricum 338 19RP3L2-7 can synthesize nitrate reductase, an enzyme essential for nitrogen assimilation by plants.” – I am not happy with this sentence. This enzyme converts nitrate into NH4, while in most species nitrogen is assimilated in the form of nitrate. I am not sure how it is with cotton
Author Response
Comments and Suggestions for Authors
This is an interesting and useful article showing capacity of some bacterial inoculants to mitigate effects of soil compaction on the growth and accumulation of mineral elements in cotton. Still I have some remarks. They should be addressed before I can recommend publishing this article.
Response: First, we must express our appreciation to you for the detailed feedback on our manuscript. We believe that the feedback has resulted in a substantially improved paper. All changes and additions of content are highlighted in red on the manuscript: plants-3635636.
- Two bacterial strains (Exiguobacterium sibiricum strain 19RP3L2-7 and Pantoea vagans strain 7URP1-6) were used for inoculation in the present work. Since it is not specified in the result section, which strains were used for inoculation, it is can be assumed that their mixture was used for plants treatment. I strongly recommend to make this clearer and to explain, if bacterial strains were incubated separately in vitro and in which proportion the cells of these strains were combined for the seed treatment.
Response: Thank you for your detailed attention and suggestion. Please see lines 82, 135 to 143, and 189 to 193.
- “(CG = (Ds / Dsm) × 100), where CN = Compaction Level, Ds = Natural 122 Soil Density, and Dsm = Maximum Soil Compaction Density” - Should it be CG or CN? CG is not deciphered, while CN is absent from the formula.
Response: Thank you for your detailed attention and suggestion. Please see lines 120.
- Citation in the text should be arranged according to the guidelines of “Plants”. I recommend to download any recent article and to check this. It is said in the guidelines that “In the text, reference numbers should be placed in square brackets.” So in the text there should be numbers and not Family et al. in the brackets.
Response: Thank you for your attention and suggestion. Done.
- One more remark concerning arrangement of the literature list. It should be checked and names of the authors presented in the same way. In the present state it is Oliveira in one case and de Oliveira – in the other.
Response: Thank you for your attention and suggestion. Done.
- “One plant per treatment was randomly selected and then segmented into leaves, roots, and rhizosphere (substrate adhered to the roots). Surface disinfestation of each plant segment was conducted as described by Oliveira et al. (2024)” – when I read this sentence, I failed to understand why plants were sterilized before assay of CFU in rhizosphere. But then I read the article of Oliveira et al., (2024) and found that “Before root disinfestation, the substrate was removed manually and the roots were placed in saline solution (9 mL of saline solution per 1 g of roots) and agitated orbitally for 30 min at 200 rpm to displace the bacteria from the rhizosphere”. So disinfection was performed after bacteria from root rhizopshere were recovered. I believe the authors should point this out rather than relying solely on the reference to Oliveira et al, (2024).
Response: Thank you for your detailed attention and suggestion. Please see lines 163 to 169.
- Furthermore, I recommend to modify the title of section 2.4.2. by adding rhizosphere: Microbial Density in Plant Tissues and Rhizosphere.
Response: Thank you for your detailed attention and suggestion. Please see line 162.
- I failed to find exact information about microbial density either in plant tissues or rhizospeher. It is only mentioned that “for colony-forming units, the statistical assumptions were not met”. Why were methods of their determination described in such details in M & M section?
Response: We would like to clarify that, given the difficulty in meeting the statistical assumptions for the quantitative analysis of the data concerning microbial density (as previously mentioned in the text), we opted to incorporate a qualitative analysis of these observations into the results.
We understand the importance of quantification wherever possible; however, the characteristics of the data collected for colony-forming units (CFUs) did not allow for a robust statistical analysis. Therefore, we have included in the text a description of the trends and patterns observed in a qualitative manner, aiming to provide relevant insights into microbial behaviour within the limitations encountered.
We believe that this qualitative approach, although not quantitative, enriches the discussion and contributes to the understanding of the investigated phenomena.
- Root characteristics under different level of soil compaction and inoculation are also not shown. Deficiency of these data is possibly due to the observation that “for root dry matter … the statistical assumptions were not met, or no significant influence of bacterial inoculation or soil compaction was observed.” Still I recommend providing this data as supplementary material
Response: Thank you for your detailed attention and suggestion. Please look at the supplementary material - plants 3635636. Table 4.
- “Shoot dry matter (SDM, g) was significantly influenced by the interaction between PGPB inoculation and soil compaction levels. Inoculated plants subjected to 75%, 85%, 216 and 95% compaction levels exhibited 47% lower SDM accumulation than at the optimal 217 compaction of 65%.” – the statement about interaction of the factors should be supported by two-way ANOVA. Its result should be presented as a table showing F, p and Fcrit values.
Response: Thank you for your detailed attention and suggestion. Please look at the supplementary material - plants 3635636. Table 1, 2, and 3.
- I recommend to start Discussion from results obtained in the present work and then to relate them to the literature data. Otherwise too many literature data are discussed, which are not related to what was measured in the present work. For example authors mention that “This effect may be related to reduced exploration of the cotton plant's root system, resulting in decreased water and nutrient absorption.” But no data on root dimensions are shown in the present work.
Response: Organizar discussão, talvez
- Throughout the text authors mention “treatments inoculated with bacteria”. I think it should not be forgotten that not treatments, but plants are inoculated. I recommend modifying such phrases throughout the text.
Response: Thank you for your detailed attention and suggestion. Done.
- “The strain Exiguobacterium sibiricum 338 19RP3L2-7 can synthesize nitrate reductase, an enzyme essential for nitrogen assimilation by plants.” – I am not happy with this sentence. This enzyme converts nitrate into NH4, while in most species nitrogen is assimilated in the form of nitrate. I am not sure how it is with cotton.
Response: Thank you for your detailed attention and suggestion. Done.

Reviewer 2 Report
Comments and Suggestions for Authors
The manuscript with the title “Potential of Bacterial Inoculants to Mitigate Soil Compaction Effects on Gossypium hirsutum Growth” is well written and interesting. Compaction is indeed a serious issue for assuring crop yield.
Here are some recommendations for the improvement of the manuscript.
Please add 2 more keywords after the abstract for example morphological parameters, inoculants
The introduction, abstract, results and discussions are well written.
Please rewrite the conclusion section and refer to all analysed parameters.
Author Response
Comments and Suggestions for Authors
The manuscript with the title “Potential of Bacterial Inoculants to Mitigate Soil Compaction Effects on Gossypium hirsutum Growth” is well written and interesting. Compaction is indeed a serious issue for assuring crop yield.
Here are some recommendations for the improvement of the manuscript.
- Please add 2 more keywords after the abstract for example morphological parameters, inoculants
Response: Thank you for your detailed attention and suggestion. Please see lines 28-29.
- The introduction, abstract, results and discussions are well written.
Response: Thank you for your detailed attention.
- Please rewrite the conclusion section and refer to all parameters analyzed.
Response: We sincerely thank the reviewer for the valuable suggestions. Done.

Reviewer 3 Report
Comments and Suggestions for Authors
This paper investigates the effects of plant growth-promoting bacteria (PGPB) on Gossypium hirsutum growth, demonstrating certain innovative aspects in its research approach. However, it should be noted that numerous studies have already reported on PGPB-mediated plant growth promotion, with many underlying mechanisms well elucidated. The current study is limited to basic measurements of Gossypium hirsutum growth parameters, presenting relatively superficial content that lacks in-depth exploration of relevant mechanisms. The authors are advised to strengthen their research in the following aspects:
- Construction of composite bacterial strains and investigation of their growth-promotion mechanisms.
- Effects of PGPB on root exudates in crops.
- Mechanisms by which PGPB enhance crop resistance to abiotic stress.
- Colonization patterns of the bacterial strains.
Author Response
Comments and Suggestions for Authors
This paper investigates the effects of plant growth-promoting bacteria (PGPB) on Gossypium hirsutum growth, demonstrating certain innovative aspects in its research approach. However, it should be noted that numerous studies have already reported on PGPB-mediated plant growth promotion, with many underlying mechanisms well elucidated. The current study is limited to basic measurements of Gossypium hirsutum growth parameters, presenting relatively superficial content that lacks in-depth exploration of relevant mechanisms.
The authors are advised to strengthen their research in the following aspects:
- Construction of composite bacterial strains and investigation of their growth-promotion mechanisms. 2. Effects of PGPB on root exudates in crops. 3. Mechanisms by which PGPB enhance crop resistance to abiotic stress. 4. Colonization patterns of the bacterial strains.
Response: We sincerely thank the reviewer for the valuable suggestions. All comments have been carefully considered, and the discussion section has been restructured based on the points raised. We believe these improvements have significantly enhanced the manuscript.

Round 2
Reviewer 3 Report
Comments and Suggestions for Authors
Accept